# Improving genetic risk prediction across diverse population by disentangling ancestry representations

Prashnna K. Gyawali [1✉], Yann Le Guen [1,2], Xiaoxia Liu[1], Michael E. Belloy[1], Hua Tang [3], James Zou [4✉] & Zihuai He [1,5✉]

Risk prediction models using genetic data have seen increasing traction in genomics. However, most of the polygenic risk models were developed using data from participants with similar (mostly European) ancestry. This can lead to biases in the risk predictors resulting in poor generalization when applied to minority populations and admixed individuals such as African Americans. To address this issue, largely due to the prediction models being biased by the underlying population structure, we propose a deep-learning framework that leverages data from diverse population and disentangles ancestry from the phenotype-relevant information in its representation. The ancestry disentangled representation can be used to build risk predictors that perform better across minority populations. We applied the proposed method to the analysis of Alzheimer's disease genetics. Comparing with standard linear and nonlinear risk prediction methods, the proposed method substantially improves risk prediction in minority populations, including admixed individuals, without needing self-reported ancestry information.

[1] Department of Neurology and Neurological Sciences, Stanford University, Stanford, CA, USA. [2] Institut du Cerveau-Paris Brain Institute-ICM, Paris, France. [3] Department of Genetics, Stanford University, Stanford, CA, USA. [4] Department of Biomedical Data Science, Stanford University, Stanford, CA, USA. [5] Quantitative Sciences Unit, Department of Medicine (Biomedical Informatics Research), Stanford University, Stanford, CA, USA. ✉email: pgyawali@stanford.edu; jamesz@stanford.edu; zihuai@stanford.edu

Prediction of complex phenotypic traits, particularly for complex diseases like Alzheimer's disease (AD) in humans, has seen increased traction in genomics research[1]. Over the past decade, genome-wide association studies (GWAS) have dominated genetic research for complex diseases like AD. Different approaches[2–4] like polygenic risk score (PRS) and wide-range of linear models have been proposed for risk prediction of complex diseases based on the genotype–phenotype associations for variants identified by GWAS. More recently, with increased data availability, non-linear methods like deep learning[5] have been considered for constructing prediction models[6].

Most genetic studies of complex human traits have been undertaken in homogeneous populations from the same ancestry group[7], with the majority of studies focusing on European ancestry. For instance, despite African American individuals being twice as likely to develop AD compared to Europeans, genetic studies on African Americans are scarce[8]. Moreover, the genomic studies for admixed individuals are limited, although they make up more than one-third of the US population, with the population becoming increasingly mixed over time[9]. This lack of representation for minority populations and admixed individuals, if not mitigated, will limit our understanding of true genotype–phenotype associations and subsequently the development of genetic risk predictions. This will eventually hinder the long-awaited promise to develop precision medicine.

The discussion for mitigating the limited diversity in genetic studies has been more pronounced recently due to the consistent observation that the existing models have far greater predictive value in individuals of European descent than of other ancestries[10]. For example, PRS for blood pressure, constructed using GWAS of European ancestry, didn't generalize well for Hispanics/Latinos[11]. Similarly, the PAGE study found that genetic risk prediction models derived from European GWAS are unreliable when applied to other ethnic groups[12]. This lack of generalization to minority populations, including admixed individuals are critical limitation of human genetics. One major reason for this poor generalizability is the prediction models being biased by the underlying population structure. Both linear and non-linear models are susceptible to overfitting the training participants, which essentially comprises European ancestry in genomics. Although the non-linear models via deep learning produce impressive results across domains, they are more prone to overfitting—failing to generalize even with minor shifts in the training paradigm[13].

Studies have shown that one way to address this is by having multi-ethnic training participants. For instance, limited generalization of PRS for blood pressure, as reviewed above, substantially increased when the GWAS of Europeans were combined with the GWAS of Hispanics/Latinos[11]. Similarly, Martin et al. pointed out that the misdiagnoses of multiple individuals with African ancestry would have been corrected with the inclusion of even a small number of African Americans[14]. Although efforts to increase the non-European proportion of GWAS participants are being implemented, the proportion of individuals with African and Hispanic/Latino ancestry in GWAS has remained essentially unchanged[15]. As such, increasing training participants of other ancestries, particularly for admixed individuals, is not likely to occur any time soon without a dramatic priority shift, given the current imbalance and stalled diversifying progress over the recent years[10].

In this work, we aim to learn robust risk prediction models that generalize across different ancestry. Previous studies for learning robust and unbiased genotype-phenotype relationships against population bias have mostly been carried out with the PRS and its variants[16,17]. These studies are focused on specific ancestry or with an assumption that at least part of the background of the genome is still of European origin[17]. In addition, PRS won't capture the complex genotype-phenotype relationship. Alternatively, deep learning has recently shown improved predictability across domains (e.g., vision, language, etc.) due to its ability to capture the complex input-output relationship. However, standard deep-learning approaches often fail to learn robust and unbiased representation, which is also the case for PRS-based methods. The major line of work for learning robust and unbiased representation with deep learning involves learning domain-invariant representation, where participants from different domains share common traits (e.g., genotype participants from different ancestry with common phenotype)[18,19]. Adversarial learning paradigms are often considered to learn such domain-invariant representations. However, such an adversarial approach is difficult to train and would require extensive hyperparameter tuning.

Here, we introduce DisPred, a deep-learning-based framework that can integrate data from diverse populations to improve the generalizability of genetic risk prediction. The proposed method combines a disentangling approach to separate the effect of ancestry from the phenotype-specific representation, and an ensemble modeling approach to combine the predictions from disentangled latent representation and original data. Unlike PRS-based methods, DisPred captures non-linear genotype-phenotype relationships without restricting specific ancestral composition. Unlike adversarial learning-based methods, DisPred explicitly removes the ancestral effect from phenotype-specific representation and involves minimal hyperparameter tuning. Although there have been recent efforts in using deep learning to capture nonlinearity within high-dimensional genomic data[6,20,21], these works haven't considered any specific strategy to separate the effect of ancestry from phenotype-specific representation. Moreover, DisPred does not require self-reported ancestry information for predicting future individuals, making it suitable for practical use because human genetics literature has often questioned the definition and the use of an individual's ancestry[22,23], and at the minimum, the ancestry information may not be available during the test time. We evaluate DisPred performance to predict AD risk prediction in a multi-ethnic cohort composed of AD cases and controls and show that DisPred performs better than existing models in minority populations, particularly for admixed individuals.

## Results

**Overview of the proposed workflow with DisPred.** DisPred is a three-stage method to improve the phenotype prediction from the genotype dosage data (each feature has a value between 0 and 2). We present the workflow summary in Fig. 1. First, as shown in Fig. 1a, we built a disentangling autoencoder, a deep-learning-based autoencoder, to learn phenotype-specific representations. The proposed deep-learning architecture involves separating latent representation into ancestry-specific representation and phenotype-specific representation. In this way, this stage explicitly separates the ancestral effect from phenotype-specific representation. Second, as shown in Fig. 1b, we used the learned phenotype-specific representation extracted from the disentangling autoencoder to train the prediction model. We consider a linear model for the phenotype prediction model in our case. Since the phenotype-specific representation is non-linear, the prediction model, despite being linear, will still capture the non-linear genotype-phenotype relationship. Finally, as shown in Fig. 1c, we create ensemble models by combining the predictions from the learned representations, i.e., the result of our second stage, with the predictions from the original data, i.e., the existing approach of building prediction models. The second stage builds the prediction model from the disentangled representations and the ensemble in the third stage aims to enhance the prediction accuracy.

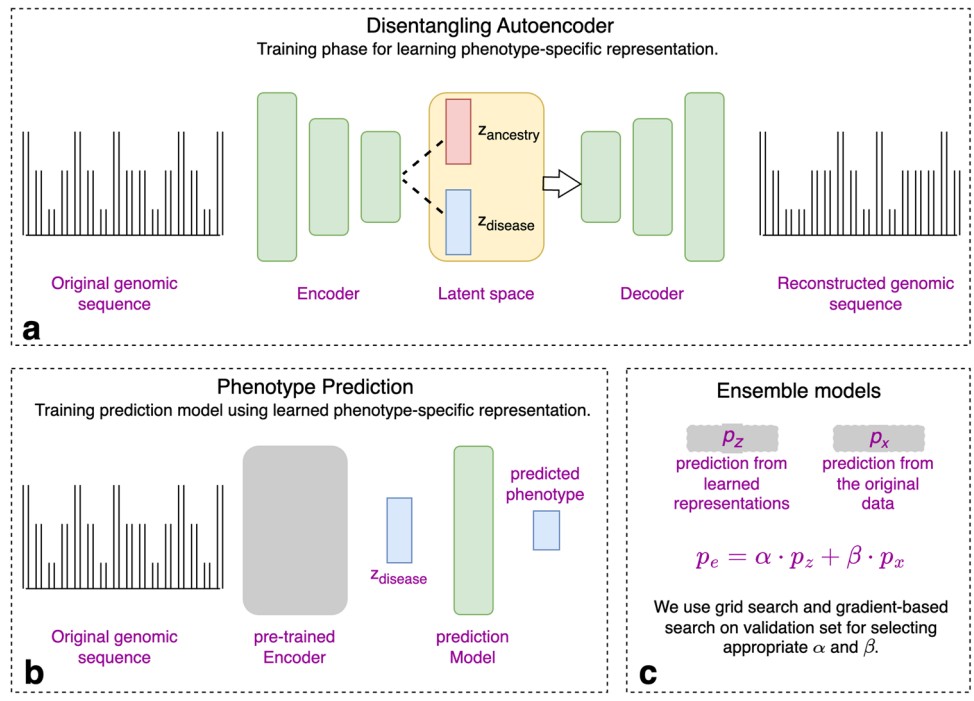

**Fig. 1 Modeling strategy. a** Disentangling autoencoder learns ancestry and phenotype-specific representation separately using autoencoder architecture and unique contrastive latent loss. **b** The disentangled phenotype-specific representation is then used for the phenotype prediction. A separate linear prediction model is trained on the obtained representation for the phenotype prediction. **c** To increase prediction power, we use the ensemble modeling approach, where the parameters can be obtained either from grid-search or gradient-based search.

We consider a training data $\mathscr{D} = \{\mathbf{x_i}, \mathbf{y_i}, \mathbf{a_i}\}_{i=1}^N$ with $N$ participants, where **y** represents the disease label (e.g., case and control for binarized data), and **a** represents the environment label for the data (e.g., categorical ancestral label). An encoder function $\mathscr{F}_\theta(\mathbf{x})$ decomposes original data **x** into ancestry-specific representation $\mathbf{z_a}$ and phenotype-specific representation $\mathbf{z_d}$, and a decoder function $\mathscr{G}_{\theta'}(\mathbf{z_a}, \mathbf{z_d})$ reconstruct the original data as $\hat{\mathbf{x}}$ using $\mathbf{z_a}$ and $\mathbf{z_d}$. Both the encoder function (θ) and decoder function (θ′) are parameterized by deep neural networks, and together represent the disentangling autoencoder. Primarily, the parameters θ and θ′ are optimized by minimizing $\mathscr{L}^{Recon}$, the average reconstruction loss over $N$ training examples.

To disentangle the latent bottleneck representation, we propose a latent loss based on the following assumptions: for any data pair originating from the same environment, the corresponding pair of latent variables $\mathbf{z_a}$ should be similar, and different if the data pair belongs to a different environment **a**. Similarly, the latent $\mathbf{z_d}$ should be similar or different if the pair belongs to the same or different disease label **y**. We propose $\mathscr{L}^{SC}$, contrastive loss[24–26] to enforce these similarities in the latent space. We apply $\mathscr{L}^{SC}$ independently to both $\mathbf{z_a}$ and $\mathbf{z_d}$. Overall, the objective function for training the disentangled autoencoder takes the following form:

$$\mathscr{L}^{Disentgl-AE} = \mathscr{L}^{Recon} + \alpha_d \cdot \mathscr{L}^{SC}_{z_d} + \alpha_a \cdot \mathscr{L}^{SC}_{z_a} \quad (1)$$

where $\alpha_*$ represent the hyperparameter for the corresponding latent loss. For each latent variable, the contrastive loss will enforce the encoder to give closely aligned representations to all the entries from the same label in the given batch encouraging disentanglement of disease and environment features in two separate latent variables.

In the second stage, we utilize the trained autoencoder to extract phenotype-specific representation $\mathbf{z_d}$ to train the prediction model for the given phenotype. Our prediction models are trained with linear regression, regressing $\mathbf{z_d}$ to the corresponding disease label **y**. Finally, in the third stage, we create ensemble

models by combining $p_z$, the predictions from the linear model using the phenotype-specific representation $\mathbf{z_d}$, with $p_x$, the predictions from the linear model using the original data **x**:

$$p_e = \alpha \cdot p_z + \beta \cdot p_x \quad (2)$$

where $\alpha$ and $\beta$ are the weighing parameters determined using gradient-based search using the validation set. We use the same training and validation data as in earlier stages to obtain $p_z$ and $p_x$ and learn the weighing parameters $\alpha$ and $\beta$. Here $p_e$ represents prediction from the additive ensemble model, a complex model that combines the predictions resulting from linear and non-linear relationship between original data and the phenotype target.

**Application of DisPred to Alzheimer's Disease (AD) risk prediction.** We use the DisPred framework to predict AD using genetic data. The aim is to evaluate the prediction accuracy (via Area under the curve: AUC) of the proposed DisPred in comparison to other methods in genomics trained on participants of European ancestry, including Polygenic Risk Score (PRS)[27], and the supervised Neural Network (NN). We considered two conventional PRS methods: PRS by clumping (PRS-Clumping) and PRS by Lasso-based penalized regression (PRS-Lasso). For all comparison methods, we considered the same set of genetic variants identified by existing GWAS as features, including 5,014 variants associated with AD from Jansen et al. (2019)[28] (variants with $p < 1e-5$) and 78 variants from Andrews et al. (2020)[29]. We evaluated these methods using two cohorts: the Alzheimer's Disease Sequencing Project (ADSP) and the UK Biobank (UKB). The outcome or phenotype label for the ADSP is the clinical diagnosis for the presence or absence of AD, and for the UKB is a dichotomized version of continuous proxy-phenotype derived from the information from family history (first-degree relative with reported AD or dementia). After filtering for participants and variants, the final dataset includes 11,640 participants for the ADSP cohort and 461,579 participants for the UKB, with 3892

variants. We split the ADSP data into training and test and divided the training set into training and validation sets with stratification based on the phenotype labels. To compare prediction accuracy, we present the results of the models tested on independent test data from the ADSP and all data from the UKB.

We consider self-reported ancestry labels in the training dataset to train the disentangling autoencoder and other ancestry-specific linear and non-linear models. The independent test data consists of 2101 participants for the ADSP and 461,579 participants for the UKB. To evaluate the performance of the proposed method to predict AD status in minority populations or admixed individuals, we estimated ancestry percentages from genome-wide data using SNPWeight v2.1[30]. Then we used these ancestry estimates to divide the test data into different ancestries to present the results. This is to ensure that the partitions accurately reflect the actual genetic-ancestry background and enable a more rigorous evaluation of the methods. Details for dataset preparation are explained in the Methods section.

Although unconventional, we also consider the models trained using other non-European ancestries like African American population (AFR) to understand the effect of training models with training participants other than European ancestry. Furthermore, we also considered adversarial learning to capture domain-invariant representations where we treated ancestry as domains (Adv). These different results are presented in the Methods section.

**Representation learned via disentangling autoencoder**. We analyzed the representations learned from the proposed DisPred framework. In Fig. 2, we present the Uniform Manifold Approximation and Projection (UMAP)[31] plots for the ancestry-specific representation $z_a$ and phenotype-specific representation $z_d$ to

assess whether the proposed method can separate the ancestry effect from the GWAS variants. First, we note that the representation captured the ancestry-related information on the left plots. The three training ancestry groups for the ADSP cohort were clearly separated and three out of four training ancestry groups for the UKB cohort were also separated. However, the admixed (MIX) group in the UKB cohort is mixed with other ancestries (left panel on bottom row). This is a limitation of the proposed regularization, or the proposed architecture, which may not be able to completely separate the MIX group from other ancestries. On the right side of both figures, when ancestry labels are applied to the phenotype-specific representation $z_d$, we note that the ancestry labels are scattered without forming clear clusters. This means that our proposed method correctly identified a latent representation $z_d$ that is invariant to ancestry background, which we later use to build the prediction models.

**DisPred improved AD risk prediction for minority populations and for admixed individuals**. We report the main results in Fig. 3. First, test data distribution for different ancestries for the two cohorts is presented in panel A. We considered 90% and 65% as estimated ancestry cut-offs, respectively, for the ADSP and the UKB to stratify test participants into five super populations: South Asians (SAS), East Asians (EAS), Americans (AMR), Africans (AFR), and Europeans (EUR), and an admixed group composed of individuals not passing cut-off in any ancestry. Since UKB primarily comprises European participants, we set the threshold low for non-EUR participants for our analysis. In both datasets, individuals of EUR ancestry correspond to the largest proportion (48% for the ADSP and 98% for the UKB). For the ADSP, we consider two sub-groups of test datasets: (i) African

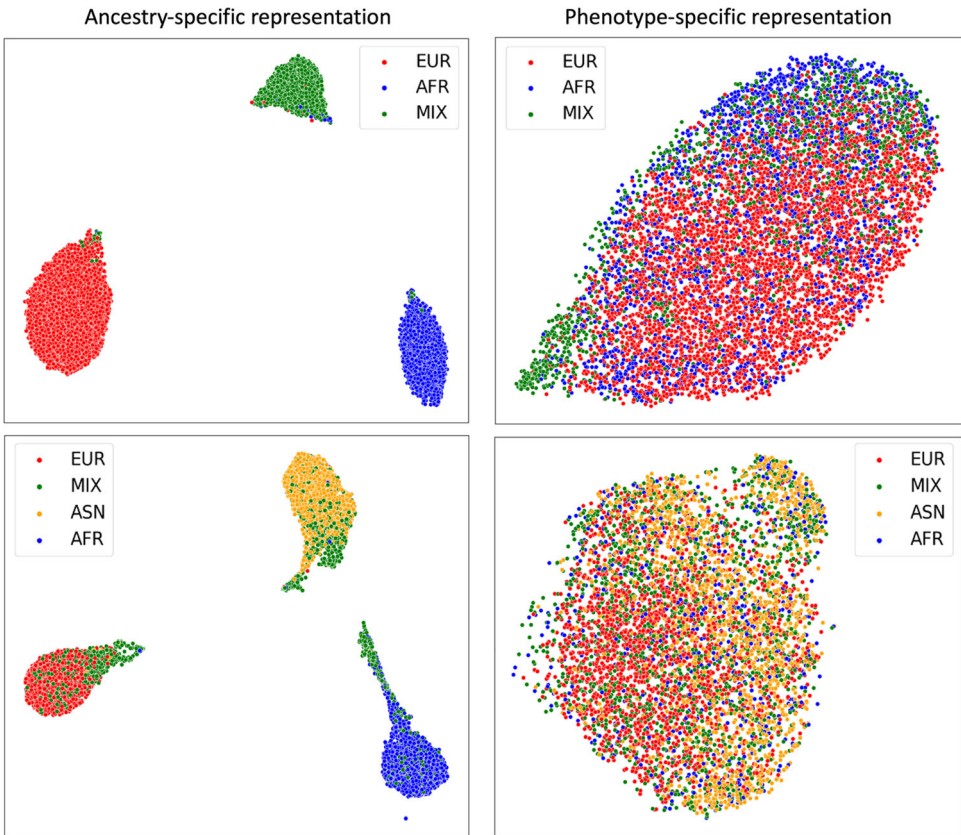

**Fig. 2 Learned latent representation.** UMAP plots of ancestry-specific representation (left column) and phenotype-specific representation (right column) were learned from Disentgl-AE for the ADSP (top row) and the UKB (bottom row). We colored the points with self-reported ancestry labels (EUR: European, MIX: admixed, ASN: Asian, and AFR: African), demonstrating that phenotype-specific representation is invariant to the ancestry background.

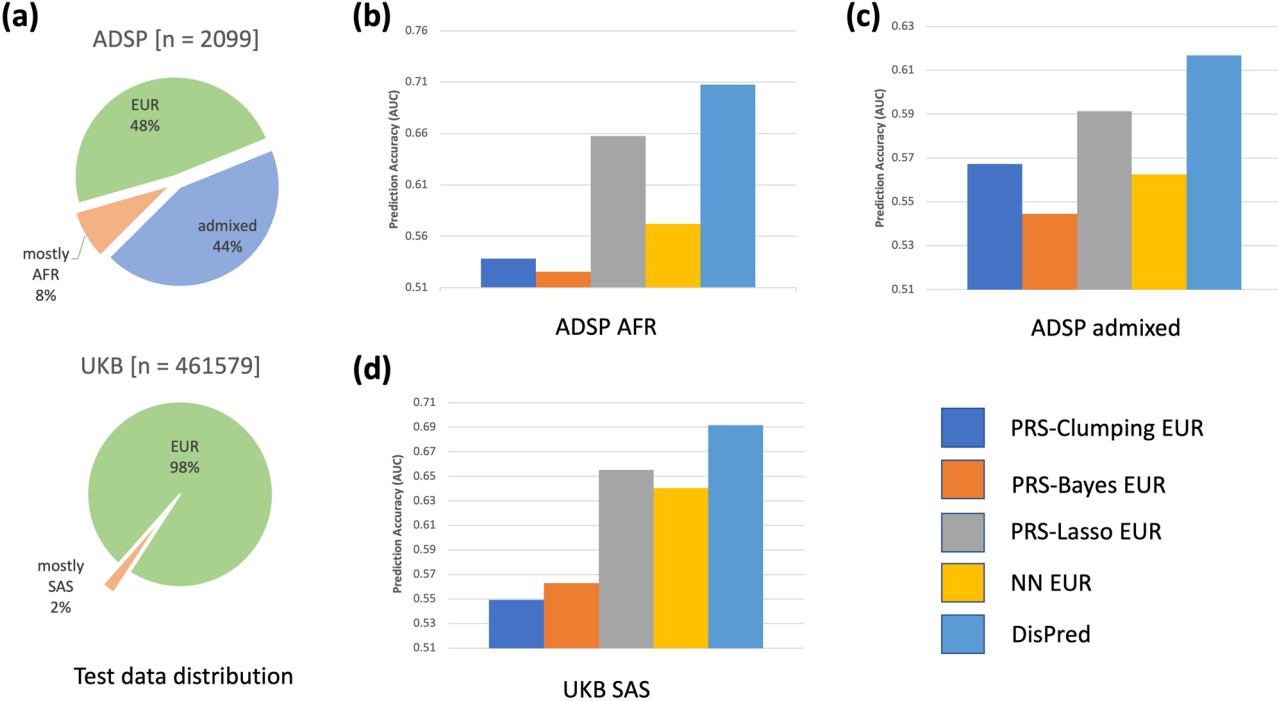

**Fig. 3 Predictive analysis. a** Test data distribution for the ADSP and the UKB cohorts. The pie chart presents the proportion of different ancestries for the ADSP and the UKB. **b** Prediction accuracy (AUC) for different models trained on European participants (PRS-Clumping, PRS-Bayes, PRS-Lasso, Neural Network (NN)) and proposed DisPred trained on all data on different subsets of the test dataset: African participants (ADSP AFR), and **c** admixed individuals from ADSP cohort, and **d** South Asian participants (UKB SAS) from the UKB cohort.

participants ($n = 169$) (ADSP AFR) (Panel B), and (ii) admixed individuals ($n = 916$) (ADSP admixed) (Panel C). This is to evaluate different models' performance on non-European minority populations. For the UKB, we consider a non-European subgroup of test datasets: (i) South Asian participants ($n = 8083$) (UKB SAS) (Panel D. This is to evaluate different models' performance on dataset shift (or distribution shift), i.e., the application of models trained on one cohort to participants of a separate cohort. Across all cases, DisPred achieves the best result. The PRS-Lasso is the second-best model in all cases, demonstrating its superiority in sparse domains above standard Neural Network models. PRS-Clumping and PRS-Bayes generally performed worst across all cases. We also note that although we did not leverage data from SAS, we significantly improved prediction accuracy for UKB SAS. The proposed Disentgl-AE utilizes all the available training participants to separate the effect of ancestry from the phenotype representation. As such, the obtained phenotype representation demonstrated the best predictive abilities.

**DisPred performs better in the presence of ancestral mismatch.** Since existing methods are often ancestry-specific, ancestry information is required for predicting future patients or individuals in a practical setting. However, human genetics literature has often questioned the definition and the use of an individual's ancestry[22,23], and at the minimum, the ancestry information may not be available during the test time. Without accurate ancestry information, the participants could either align or mismatch with the corresponding ancestry of training populations. For such conditions, we present the results in Fig. 4. We consider two different test scenarios: (i) ADSP EUR ($n = 1014$) and (ii) ADSP AFR ($n = 161$) and compared the performance of DisPred against models trained with ADSP EUR participants (left panel) and models trained with ADSP AFR participants (right panel). We use the same ancestry percentage cut-off in the previous section to stratify the test participants into EUR and AFR for ADSP.

Since the effect sizes (or betas) for calculating PRS are derived from EUR participants, we don't have results for PRS-Clumping AFR in the above table. In each panel, the dotted lines separate the scenarios for ancestral alignment (left) or mismatch (right). We note that among all the analyzed cases, only when the EUR test participants aligned with the EUR training population, DisPred obtain less predictive accuracy than PRS-Lasso. Other than that, DisPred achieved better results when the test participants didn't align with the training population for AFR and EUR analysis and for AFR, even when they aligned. This demonstrates that we learned an invariant representation robust to ancestry background. Overall, these results show the practical usability of the DisPred model over existing methods when there is an ancestral mismatch.

**DisPred with increasing individual-level heterogeneity.** DisPred, compared to existing methods, improved predictability for the minority population, including admixed individuals. Since admixed individuals' genome is an admixture of genomes from more than one ancestral population, we identified that different models struggle as individual heterogeneity increases. This section presents an in-depth evaluation of this issue for the ADSP cohort. Estimated ancestral percentages were used to calculate the heterogeneity. For each sample in the test set, the variance of ancestral proportion was computed, and we sorted these in decreasing order, obtaining a sequence from homogeneity to heterogeneity. For such an arrangement, Fig. 5 shows the results, where we created numerous data subsets by sliding window-based process with a window size of 750 participants and a stride length of 50 participants. This way, the data subset heterogeneity increases from left to right. The background is color-coded with the proportion of the estimated ancestral percentage for each data subset. All methods considered in this work start to degrade as we move toward heterogeneity or admixed individuals. The proposed DisPred performs well when there is a sharp decline in the

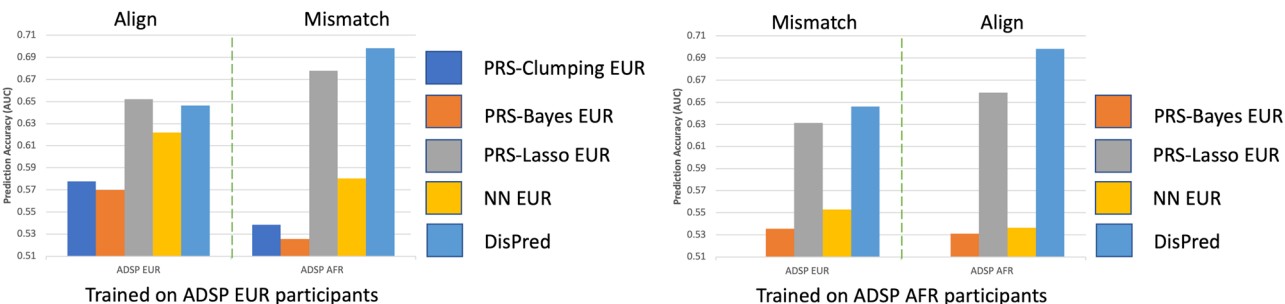

**Fig. 4 Predictive performance for misaligned train and test cohorts.** Prediction accuracy (AUC) for different models (PRS-Clumping trained on European (PRS-Clumping EUR), PRS-Bayes trained on European (PRS-Bayes EUR), PRS-Lasso trained on European (PRS-Lasso EUR), PRS-Bayes trained on African (PRS-Bayes AFR), PRS-Lasso trained on African (PRS-Lasso AFR), Neural Network trained on European (NN-EUR), Neural Network trained on African (NN-AFR), and DisPred) on the practical setting of when there is alignment or mismatch between the ancestry of training and test participants. All prediction models are trained on ADSP (EUR participants on the left and AFR participants on the right).

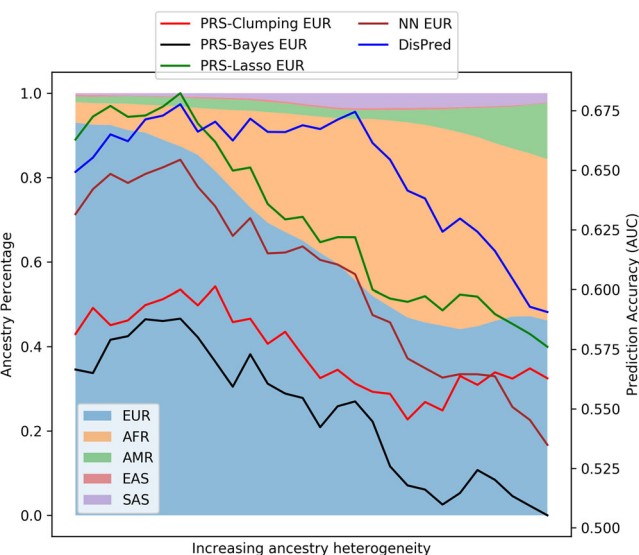

**Fig. 5 Predictive performance with increasing ancestry heterogeneity.** Prediction accuracy (AUC) for different models on data subsets with increasing ancestry heterogeneity for the ADSP cohort. The background is color-coded with the proportion of the estimated ancestral percentage (EUR: European, AFR: African, AMR: American, EAS: East Asian, and SAS: South Asian) for each data subset.

proportion of European ancestry (in the middle region of the graph) and slowly decreases towards the end. Overall, the DisPred produces the best result. These results also suggest that the lack of generalization cannot be addressed by simply increasing the non-European training participants by recruiting more homogeneous individuals from minority populations. An equal effort needs to be given toward designing unbiased and fairer algorithms like DisPred.

## Discussion

As the population becomes increasingly mixed over time, understanding how to analyze and interpret admixed genomes will be critical for enabling transethnic and multi-ethnic medical genetic studies and ensuring that genetics research findings are broadly applicable. Yet, existing AI-based approaches in genomics are largely focused on homogeneous European ancestry. We thus urge a joint research effort to confront the existing approach for ancestry-specific AI frameworks and focus on building unbiased alternatives. This study introduced DisPred, a deep-learning-based framework

for improving AD risk prediction in minority populations, particularly for admixed individuals. First, we developed a disentangling autoencoder to disentangle genotypes into ancestry-specific and phenotype-specific representations. We then build predictive models using only phenotype-specific representation. Finally, we used ensemble modeling to combine the prediction models built using disentangled latent representation, and the model built using the original data. We constructed DisPred for unbiased and robust predictions for diverse ancestry, particularly for the non-European population. With AD GWAS data from the ADSP and the UKB cohort, we confirmed the effectiveness of the disentanglement-based framework for non-European minority populations.

In this study, we also demonstrated that, unlike existing practices where models built for particular ancestry are applied to individuals of the same ancestry requiring ancestry information at the test time, DisPred is robust to different ancestry compositions without the need for self-reported ancestry information. We believe this is an appealing feature of DisPred as there has been a long-standing debate on the use of self-reported race/ethnicity/genetic ancestry in biomedical research. The scientific, social, and cultural considerations make it challenging to provide an optimal label for race or ethnicity, which could result in ambiguity, contributing to misdiagnosis. Moreover, at the minimum, such ancestry information may not be available. As such, a method like DisPred that does not require ancestry information is beneficial to get around this critical issue. Similarly, dataset shift, i.e., different training and test data distributions (or cohorts), although a common practical scenario, makes generalization challenging for machine learning models. We showed that compared to existing methods, DisPred demonstrated better predictive abilities in the presence of dataset shift. However, note that, compared to other methods, the increase in prediction performance for non-European individuals comes with a decrease or similar prediction performance for European individuals. Overall, existing methods typically improve predictions by leveraging data from the target population and, in turn, make the model more sensitive to the ancestry background and thus require relevant ancestry information at prediction stages. Unlike them, DisPred is different and unique as it learns invariant representation robust to ancestry background and performs well in a diverse test environment.

However, certain limitations are not well addressed in this study. First, in this study, we didn't perform any feature importance analysis to understand if specific variants were more pronounced for admixed groups than the homogenous European ancestry. Due to the multi-stage framework involving different training and optimization processes, unlike other AI approaches, it is not straightforward to relate the predicted phenotype to the original data level. A detailed study is required to trace the flow of

information from the genetic variants to disentangled representations and then to the predicted phenotype. In this study, we couldn't perform extensive comparisons with several related methods. In particular, the PRS-based multi-ethnic genetic prediction approaches[32–36] are one of the most active statistical genetic research areas and have shown improvement for non-European populations. Most of these works propose statistical methods to integrate genome-wide association study summary statistics from different populations effectively. Our work on the other hand, doesn't incorporate such summary statistics (which can be from multiple ancestries), and instead trains on individual-level data. The third is the engineering efforts. The deep-learning architecture, proposed in this work constructed using multi-layer perceptron with ReLU activation, potentially has space for improvements. Our framework used contrastive loss[24–26] to achieve disentanglement in the latent space. In recent years, many advanced similarity-enforcing losses[37,38] have been proposed in the deep-learning literature, which could improve our framework. We have open-sourced the disentangling autoencoder to encourage future research in this direction. It is worth mentioning that our proposed framework is not confined to AD risk prediction and can be extended to other phenotypes. Third, we restricted the training set to significant variants, and with PRS-clumping, this set was further shrunk. As such, it might seem a bit unfair comparison as it is known that sub-threshold single nucleotide polymorphisms (SNPs) also carry information. However, we note that all the methods are compared with the same set of variants and participants, so the overall comparison is fair. Further, the use of significant variants (threshold of 1e-5 or 5e-8) for risk calculation or similar other meta-analyses is still common in AD genetics literature[1,39].

## Methods

In this section, we first describe how we constructed the dataset and provide details of our proposed method and implementations for collaboratively training the AI model.

**Dataset preparation.** This study considers two datasets: the ADSP and the UKB. We use the $p < 1e-5$ threshold to obtain the candidate regions of 5014 GWAS variants obtained from Jansen et al. (2019)[28] and Andrews et al. (2020)[29]. We remove SNPs with more than a 10% missing rate to ensure marker quality resulting in 3892 variants. We then remove participants with absent AD phenotype or ancestral information. For the ADSP, we have a dichotomous case and control label for the AD phenotype. For the UKB, we use the AD-proxy score defined in Jansen et al. (2019)[28], which combines the self-reported parental AD status and the individual AD status. Since AD is an age-related disease, we removed control participants below age 65 (age-at-last-visit). A total of 10,504 participants were obtained for the ADSP cohort and 461,579 participants for the UKB. We held out 20% of the participants for the ADSP for the test data and the rest as the training data. The training data consists of 8403 participants, including 2061 AFR, 4780 EUR, and 1562 HIS based on self-reported ancestry labels. These self-reported ancestry labels are incorporated into the proposed method to learn the latent representation. Further, we held out 1000 participants as the validation data. The data division into training, test, and validation is stratified using phenotype labels, i.e., each set contains approximately the same percentage of participants of phenotype labels as the complete set. Overall, the number of training, test, and validation participants for the ADSP is 7403, 2101, and 1000, respectively. For the sub-groups test datasets presented in the main result (Fig. 3), the case-control statistics are as follows, ADSP

AFR ($n = 169$, case $= 63$, control $= 106$), and ADSP admixed ($n = 916$, case $= 340$, control $= 576$).

**Ancestry determination.** For each cohort included in our analysis, we first determined the ancestry of each individual with SNPWeight v2.1[30] using reference populations from the 1000 Genomes Consortium[40]. Prior to ancestry determination, variants were filtered based on genotyping rate (<95%), minor allele frequency (MAF < 1%), and Hardy–Weinberg equilibrium (HWE) in controls ($p < 1e-5$). By applying an ancestry percentage cut-off >90% (for ADSP) and >65% (for UKB), the participants were stratified into five super populations: South Asians, East Asians, Americans, Africans, and Europeans, and an admixed group composed of individuals not passing cut-off in any ancestry.

**Disentangling autoencoder.** The disentangling autoencoder comprises of an encoder function $\mathscr{F}_\theta(\cdot)$ and the decoder function $\mathscr{G}_{\theta'}(\cdot)$. The encoder decomposes original data $\mathbf{x}$ into ancestry-specific representation $\mathbf{z_a}$ and phenotype-specific representation $\mathbf{z_d}$, and the decoder reconstruct the original data as $\hat{\mathbf{x}}$ using $\mathbf{z_a}$ and $\mathbf{z_d}$, which are concatenated together. The parameters of the autoencoder, i.e., $\theta$ and $\theta'$ are optimized by minimizing the $\mathscr{L}^{Recon}$, average reconstruction error over $N$ training examples.

$$\mathscr{L}^{\text{Recon}} = \frac{1}{N}\sum_{i=1}^{N} L_r(x_i, \hat{x}_i) = \frac{1}{N}\sum_{i=1}^{N} L_r(x_i, \mathscr{G}_{\theta'}(\mathscr{F}_\theta(x_i))) \quad (3)$$

where $L_r$ is mean squared error. To achieve disentanglement, we propose contrastive loss[24–26] to enforce the similarities between the latent representations obtained from the data pair in the latent space. For a batch with $N$ randomly sampled pairs, $\{x_k, y_k\}_{k=1}^N$, the contrastive learning algorithm use two random augmentations (commonly referred to as "view") to create $2N$ pairs $\{\widetilde{x}_l, \widetilde{y}_l\}_{l=1}^{2N}$, such that $\widetilde{\mathbf{y}}_{[1:N]} = \widetilde{\mathbf{y}}_{[N+1:2N]} = \mathbf{y}_{[1:N]}$. In our case, we simply replicate the batch to create $2N$ pairs. Using this multi-viewed batch, with index $i \in I \equiv \{1\dots 2N\}$ and its augmented pair $j(i)$, the supervised contrastive (SC) loss[41] takes the following form:

$$\mathscr{L}^{SC} = -\sum_{i\in I}\frac{1}{S(i)}\sum_{s\in S(i)}\log\frac{\exp(\mathbf{z}_i \cdot \mathbf{z}_s)/\tau}{\sum_{r\in I_i}\exp(\mathbf{z}_i \cdot \mathbf{z}_r)/\tau} \quad (4)$$

where $S(i)$s $\equiv \{s \in I\backslash i : \widetilde{y}_p = \mathbf{y}_p\}$ is the set of indices of all the positives in the multi-viewed batch distinct from $i$ and $\tau$ is a temperature parameter. We apply $\mathscr{L}^{SC}$ independently to both $\mathbf{z_a}$ and $\mathbf{z_d}$. Overall, the objective function for training disentangled autoencoders takes the following form:

$$\mathscr{L}^{\text{Disentgl}-AE} = \mathscr{L}^{\text{Recon}} + \alpha_d \cdot \mathscr{L}^{SC}_{z_d} + \alpha_a \cdot \mathscr{L}^{SC}_{z_a} \quad (5)$$

where $\alpha_*$ represent the hyperparameter for the corresponding latent loss. We set these hyperparameters through grid search using a held-out validation set. For each latent variable, the contrastive loss will enforce encoder to give closely aligned representations to all the entries from the same label in the given batch encouraging disentanglement of disease and environment features in two separate latent variables.

**Model architecture and training setting.** We used the Python package scikit-learn[42] to implement all the linear models and PyTorch[43] to implement all the non-linear models, including the proposed Disentangling Autoencoder. The proposed disentangling autoencoder (Supplementary Fig. 1, top row) comprises four layers in the encoder and three layers in the decoder. The encoder layers are designed to learn low-dimensional features, and the decoder layers to upscale the low-dimensional features

into high-dimensional data space. We consider the ReLU activation function for all the non-linear layers in the autoencoder. We train DisentglAE with Adam[44] optimizer with a constant learning rate of 5e-3. We train the model till $N$ epochs. We first let the model focus entirely on the reconstruction task till $N_1$ epochs by setting the weight parameters for similarity loss as 0. From $N_1 + 1$ epochs and to $N_2$ epochs, we linearly ramp the weight parameters $\alpha_*$ to 1, and then continue training till $N$ epochs. We conduct hyperparameter testing selecting the model that results in the best validation AUC for the following hyperparameters: $z_d$ (30, 40, 50), $z_a$ (30, 40, 50), $\tau$ (0.03, 0.05), and $\alpha_*$ (0.0001, 0.0003), where $z_d$, $z_a$, $\tau$, and $\alpha_*$ are selected, respectively, as 40, 40, 0.03, and 0.0001. Other used values for the hyperparameter include, $N = 500$, $N_1 = 100$, $N_2 = 250$, and batch size $= 256$. We then use ordinary least square linear regression to minimize the residual sum of squares between $y$, the phenotype labels and $p_z$, the targets produced by the $z_d$.

For the ensemble modeling, we combine $p_z$, predictions from learned representations and $p_x$, prediction from the original data. For the prediction from the original data, we consider Lasso linear model to make fair comparison with existing models. To combine these predictions, we conduct both grid-search and gradient-based search. For grid-search, we test all the values from 0.1 to 1.5, increasing at 0.1 for both $\alpha$ and $\beta$, and found $\alpha = 1.4$ and $\beta = 0.4$ for the ADSP, and $\alpha = 1.2$ and $\beta = 0.6$ for the UKB, which produces the best AUC for the validation set. For the gradient-based search, we consider $\alpha$ and $\beta$ as the parameter, initialize their weights as 1.1 and 0.9, and train the ensemble function ($p_e$) with an SGD optimizer for 5000 epochs. The best result was produced by $\alpha = 1.103$ and $\beta = 0.720$ for the ADSP, and $\alpha = 1.121$ and $\beta = 0.560$ for the UKB. The reported results for both UKB and the ADSP using the parameters from gridsearch. The result from the gradient search was similar to the grid-search result.

We used the Sherlock high-performance computing (HPC) device at Stanford University to conduct our experiments. The configurations for the cluster of remote machines were GPU: Nvidia GeForce RTX 2080 Ti 11GB; and CPU: Intel Xeon Silver 4116 2.10 GHz; OS: Ubuntu16.04.3 LTS. The computational time for DisPred primarily depends on the number of input features, dataset size, batch size, and total epochs used to train the Disentangling autoencoder. For the standard setup in our study, using the computing resources presented above, training of Disentangling Autoencoder for 500 epochs for the ADSP training dataset took 14 min.

**Baseline methods**. In Supplementary Fig. 1 (middle row and bottom row), we present the architecture for other neural network-based methods considered in this work. This includes supervised Neural Network (NN) and adversarial learning for capturing domain-invariant representations with ancestry as domains (Adv). For training NN, we consider the following parameters for ADSP: number of epochs = 200, learning rate = 5e-3, batch size = 64, and for the UKB: number of epoch = 100, learning rate = 5e-3, batch size = 256. For Adv, we consider Wasserstein distance for capturing domain-invariant representation and followed the experimental setup from Shen et al. (2017)[45]. We provide the detailed results of these methods in Supplementary Note 1 and Supplementary Figs. 2 and 3.

To derive PRS, we used three different approaches: PRS by clumping (PRS-Clumping), PRS by Naive Bayes-based penalized regression using individual-level data (PRS-Bayes), and PRS by Lasso-based penalized regression using individual-level data (PRS-Lasso). For PRS-Clumping, we start with a PRS based on the well-known APOE locus for AD risk prediction (rs429358 and rs7412)

and then perform clumping (i.e., thinning and prioritizing associated SNPs) for the rest SNPs so that the retained SNPs are largely independent of each other[27]. To ensure independence, we used the Pearson correlation, and starting from the most significant SNPs (anchor SNP), we removed all other SNPs with $R^2$ greater than 0.5 that are within 1MB distance of the anchor SNP. We then compute the sum of risk alleles corresponding to the AD phenotype for each sample, weighted by the effect size estimates. We obtain the effect size from the genetic variants identified by Jansen et al. (2019)[28] and Andrews et al. (2020)[29]. For PRS-Bayes, we construct polygenic risk scores via Gaussian Naive Bayes using individual-level data from ADSP. The best model is selected by 5-fold cross-validation using the whole training set. Unlike PRS-Clumping and PRS-Lasso, PRS-Bayes doesn't induce any additional sparsity. For PRS-Lasso, we construct polygenic risk scores via Lasso penalized regression[46] using individual-level data from ADSP. The Lasso linear models are fitted with iterative fitting along a regularization path, and the best model is selected by 5-fold cross-validation using the whole training set. We set alphas automatically and consider $N_{alpha}$ (number of alphas along the regularization path) = 10, the maximum number of iterations = 5000, and the tolerance value for optimization = 1e-3.

**Reporting summary**. Further information on research design is available in the Nature Portfolio Reporting Summary linked to this article.

## Data availability

The dataset used in this paper, i.e., the Alzheimer's Disease Sequencing Project (ADSP) and the UK Biobank (UKB), are publicly available data cohorts and are available at https://adsp.niagads.org and https://www.ukbiobank.ac.uk, respectively. We used publicly available summary statistics from Jansen et al. (2019)[28] and Andrews et al. (2020)[29]. The source data behind the graphs in Figs. 3, 4 and 5, and Supplementary Figs. 2-3 can be found in Supplementary Data 1, 2, 3, and 4, respectively.

## Code availability

The code for data preprocessing is written in R programming language. The code for DisPred's training, prediction and evaluation are written in Python with PyTorch and scikit-learn. The codes are available on the GitHub platform (https://github.com/Prasanna1991/DisPred)[47].

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

## Acknowledgements
This work is supported by NIH/NIA award AG066206 (Z.H.).

## Author contributions
P.K.G., J.Z., and Z.H. developed the concepts for the manuscript and proposed the method. P.K.G, Y.L.G., X.L., M.E.B., J.Z., and Z.H. designed the analyses and applications and discussed the results. P.K.G, J.Z., and Z.H. conducted the analyses. Z.H., J.Z., and H.T. helped interpret the obtained results. P.K.G., Z.H., J.Z., Y.L.G., X.L., and H.T. prepared the manuscript and contributed to editing the paper.

## Competing interests
The authors declare no competing interests.
