## [Peer Review File · Communications Biology]

Reviewers' comments:

Reviewer #1 (Remarks to the Author):

In this manuscript, Gyawali et al. proposed a machine learning based method, DisPred, to predict the polygenic risk scores (PRS) particularly for diverse populations. The method consists of three stages, disentangling autoencoder, phenotype prediction and ensemble. They applied the proposed method to Alzheimer's disease to show the advantage of DisPred. The method is innovative but I have a few questions and comments.

1. The authors only compared DisPred with some methods/models that were not commonly used in the PRS research. They didn't mention some of the state-of-the-art PRS methods. It's unclear which criteria the authors used to select methods to compare with.

2. There are some issues in the real data applications, making it hard to judge the advantage of DisPred.

(1) It is unclear to me how the authors exactly applied DisPred in their AD applications. How many ancestries did the authors consider in the disentangling autoencoder framework?

(2) More PRS methods should be added for comparison, for example, P+T/C+T, PRS-CSx, etc..

(3) The authors only "considered genetic variants identified by existing GWAS as features, including 5,014 variants associated with AD from Jansen et al., 2019 (variants with $p < 1e-5$) and 78 variants from Andrews et al., 2020". Why including only so few variants? The authors aimed to capture non-linear relationships, but some non-linear relationships may not be detected in a standard GWAS. Also, other methods can perform efficiently on a much looser p-value threshold with including much more variants. Will DisPred still be favorable?

(4) Jensen et al. 2019 included UKB participants in their discovery GWAS, while the authors also included the same individuals in their PRS testing, which will cause over-estimate the PRS performance. Discovery cohorts, PRS training cohort and PRS testing cohort should be completely independent for better evaluations.

(5) The sample size for non-EUR participants in ADSP is only 169 (shown in Figure 3E), which will make the results unstable. How many cases are there in the different sets of individuals?

(6) It seems all the results were based on only one set of results. The authors should consider multiple repeats to obtain reliable comparisons. Is DisPred robust to different random seeds, different ancestral admixtures?

3. No computational costs and codes availability mentioned about DisPred.

Some minor issues:

1. Some literature reviews about previous efforts of deep learning models in polygenic scores would be helpful.

2. Figure 2 colors by self-reported ancestry, but it may not correctly reflect the genetic ancestry as the lower left panel shows. It will be better to color by genetic ancestry. Also, it seems there are still some clusters based on the colors in the phenotype-specific representations. For example, most of the red dots are on the left and yellow dots on the right in the lower right panel. Any reasonable explanations?

3. Different ancestry cutoffs for ADSP and UKB doesn't make sense to me. The authors should consider using the same thresholds.

4. Figure 4 is not informative. It seems the performance of DisPred on the two panels of the figure are exactly the same. More explanations needed.

Reviewer #2 (Remarks to the Author):

The authors propose a deep learning approach for genetic risk prediction to improve generalizability across different ancestries. DisPred is a three-staged approach consisting of a disentangling autoencoder to split ancestral and phenotype-specific information, a linear prediction model and an

ensemble model. They illustrate their approach on the example of Alzheimer's disease (AD) using data from the Alzheimer's disease Sequencing Project (ADSP) and the UK Biobank (UKB). The method is compared to two more classical PRS (via Clumping and via Lasso) and other deep learning approaches (one supervised neural network and two adversarial learning models) and generalizes better across different ancestries in the presented dataset.

The transferability of genetic risk scores among different populations is a hot topic in biomedical research. The authors propose a novel approach to this field with promising results and the manuscript is well written. However, further comparisons are required to strengthen the results of the paper.

I have the following major comments:

1. The authors compare DisPred to two classical PRS approaches (Clumping and Lasso). However, they do not use any additional information/covariates in the training of those methods whereas they provide ancestral information in the training of DisPred. In the PRS field, it is common to include the first (4-10) principal components of the genotype matrix in the regression model to account for different ancestries. Furthermore, calculating principal components would also not need self-reported ancestries (similar to the autoencoder). I would suggest including principal components in the compared approaches to provide a fairer comparison.
2. The two compared PRS methods (Clumping and Lasso) both induce sparsity to the PRS model (even though the preselected variant set is sparse compared to the complete number of SNPs). Can the authors include a method that does not induce additional sparsity as comparison (e.g. SBayesR or PRScs)?
3. The authors themselves mention that the training set is strongly restricted to significant variants. While I agree that it's fair that all methods are based on the same training data, this is not a realistic approach to compute PRS. Most methods, either based on summary statistics as SBayesR, LDpred2 or lassosum or individual-level-data based as snpnet (Lasso), would be based on genome-wide data. Can the authors add additional comparisons based on genome-wide data (also for DisPred, if possible)?
4. It would be helpful to show confidence intervals of the AUC (e.g. bootstrapped CI) in Figure 3 to assess the significance of the differences.
5. In Figure 4, it seems that both models, DisPred and PRS-Lasso EUR, perform worse on the target population (EUR) than on the mismatch population (AFR). Can the authors comment on that?
6. For European data, DisPred does not seem to be the best method, i.e. the increase in prediction performance for non-EUR individuals comes with a decrease in prediction performance for EUR individuals. I would suggest mentioning that point in the discussion.
7. In lines 188-191 it is stated that the training set consists completely of ADSP data whereas the UKB data is used for testing purposes only. However, in the methods section "Model architecture and training setting" it seems that the autoencoder was also trained on UKB data. Could the authors please clarify if a separate autoencoder was trained on UKB data?

I have the following minor comments:

1. For the prediction model, as the disease label is binary, I assume it is rather a logistic regression than a linear regression (lines 161-166).
2. In the methods section "Baseline methods" it says " PRS by clumping (PRS-Clumping) and PRS by Lasso-based penalized regression using summary statistics (PRS-Lasso)" (lines 475-476) but it seems that the Lasso was fitted on the (pre-filtered) individual-level data instead of summary statistics.

Reviewer #3 (Remarks to the Author):

In this work, Gyawali and colleagues present DisPred, a deep learning framework to improve genetic risk prediction in diverse ancestral populations. This is a clearly an important research topic and the authors did a good job describing their approach and summarizing the results. Overall this is a manuscript of good quality. I just have some comments on the literature review/comparison and method details.

Major comments:

1. AD was the featured application in this paper but the authors claim that these results can be generalized to other phenotypes without performing simulations under various types of genetic architecture. If the authors do not plan to add simulations in the revision, at least remove the APOE region (the whole region, not just the 2 SNPs) from the data and see if DisPred still achieves a substantial gain compared to other approaches?
2. Is there a reason not to adjust for covariates in the analysis? Even if there are methodological limitations that make it difficult to include covariates during model training, it should be an important to add covariates (e.g., genetic PCs) in model evaluation just to be sure that the gain we see in DisPred is not explained by confounding (e.g., with ancestry composition within each sub population).
3. The literature review in this paper is fairly inadequate. Multi-ethnic genetic prediction is one of the most active statistical genetic research topics in the past 2~3 years. Plenty of methods have been developed and published. Just to give some examples (and there are many more!):

XPASS (Cai et al 2021)
PRS-CSx (Ruan et al. 2022)
PolyPred (Weissbrod et al. 2022)
X-Wing (Miao et al. 2022)
SDPRX (Zhou et al. 2022)

First of all, it would be important to present this literature so that the readership understands that existing work is not limited to the couple of papers cited by the authors which try to leverage admixture info to improve PRS performance. Second, although I do not think a comprehensive comparison with all these existing approaches is a must-do for this journal, I am wondering what is the author's justification for it. Is it because these methods only use GWAS summary statistics as input while DisPred uses individual-level data? But then you did compare with PRS-clumping and PRS-lasso which are much less sophisticated than these other methods. I understand the amount of work that's needed for comparing with all these methods in a revision which is why I said earlier this may be optional. But I think it would be a big limitation of this paper which should be addressed in the Discussion section if the authors choose not to do so.

4. Related to the comment above, the fact that DisPred requires individual-level data will be viewed as a limitation by many in the field. It doesn't seem straightforward for DisPred to incorporate publicly available GWAS summary statistics (which can be from multiple ancestries) into its model training procedure. It will be important to mention this in the Discussion section.
5. How well does DisPred perform if P_z and P_x (Fig 1c) are evaluated separately? I'm wondering if the gain of performances comes from P_z or from the fact that multiple models are combined through ensemble learning.

6. Comments on computation time and cost will be helpful. The fact that DisPred needs to use a truncated SNP list as input is a limitation for many complex traits since previous investigations have favored genome-wide models for traits with a more "omnigenic" genetic architecture. If DisPred cannot handle genome-wide SNP data, this needs to be discussed.

Minor comments:

7. Abstract and Intro: I am not sure "confounded by the underlying population structure" is the right way to describe the PRS transferability problem. It is a common view in the field that LD and allele frequency differences, and differences in genetic causal effects (which can be caused by gene-environment interaction for example) are all plausible reasons for low portability of PRS across ancestries. Maybe you could argue MAF difference is a form of "confounding" but certainly some of these other factors aren't.

8. Consider revising the color scheme in Fig 5. It's a bit hard to relate the lines to the legend.

Response to Reviewers

We thank the reviewers for their constructive feedback. In this paper, we investigate the ability of deep neural networks, in particular, disentangled representation learning to improve genetic risk prediction across diverse populations by disentangling ancestry representations. Toward this, we propose a novel deep-learning framework that leverages data from diverse populations and disentangles ancestry from the phenotype-relevant information in its representation. The ancestry disentangled representation can be used to build risk predictors that perform better across minority populations. We applied the proposed method to the analysis of Alzheimer's disease genetics. We have made substantial revisions¹ to the manuscript to address the concerns raised during the manuscript review stage.

Below we outline our response to all the issues raised by the reviewers.

Reviewer 1

Q. The authors only compared DisPred with some methods/models that were not commonly used in the PRS research. They didn't mention some of the state-of-the-art PRS methods. It's unclear which criteria the authors used to select methods to compare with.

Response: The primary contribution of our work is the novel deep learning approach and its application in Alzheimer's disease genetics. Since some of the state-of-the-art PRS (e.g. PRS-CSx) methods consider variants at genome-wide level and the limitation of our current studies is that we haven't analyzed our model performance at such genome-wide level. However, in our revised version, we have discussed this and pointed out several state-of-the-art methods.

Q. More PRS methods should be added for comparison, for example, P+T/C+T, PRS-CSx, etc.

Response: We thank reviewers for this constructive feedback. As a result in this revised version, we have added PRS-Bayes, PRS approach constructed without adding additional sparsity. Although this method isn't exactly the same as the suggested methods, but the PRS methods in the current version (e.g., Lasso, Clumping, Bayes, etc.) captures the foundation of the methods.

Q. The authors only "considered genetic variants identified by existing GWAS as features, including 5,014 variants associated with AD from Jansen et al., 2019

¹ Most of the textual revisions made in the manuscript are colored in blue.

(variants with $p < 1e-5$) and 78 variants from Andrews et al., 2020". Why including only so few variants? The authors aimed to capture non-linear relationships, but some non-linear relationships may not be detected in a standard GWAS. Also, other methods can perform efficiently on a much looser p-value threshold with including much more variants. Will DisPred still be favorable?

Response: We agree that this is the current limitation of our study. The scope of the current work aims to propose a new deep learning approach and its application in AD genetics and thus, we focus on limited variants. However, we demonstrate that our method, compared to all the standard methods considered in the genetics literature, performs well, especially for the minority population, which essentially what the design of the deep learning approach (i.e., learning disentangled representation) aims for. As such, in future, when we have efficient neural networks that can be easily trained and executed in genome-wide level, we believe the framework like ours could still be favorable as we learn the low-dimensional non-linear structure and use such latent space to remove the underlying population structure making our prediction unbiased and fair.

Q. Jensen et al. 2019 included UKB participants in their discovery GWAS, while the authors also included the same individuals in their PRS testing, which will cause over-estimate the PRS performance. Discovery cohorts, PRS training cohort and PRS testing cohort should be completely independent for better evaluations.

Response: Although the European participants in the UKB might be overlapping and potentially overestimate in the PRS performance, the non-European part of UKB would be completely independent, and that's the primary result that we have demonstrated. To avoid any confusion regarding this aspect, we have moved our results for European UKB participants to the supplementary side in the revised manuscript.

Q. The sample size for non-EUR participants in ADSP is only 169 (shown in Figure 3E), which will make the results unstable. How many cases are there in the different sets of individuals?

Response: Although the sample size used for evaluating the model for ADSP dataset is small, the training sample size is reasonable to have stable training. The case control distribution for those test dataset is as follows: ADSP AFR (n = 169, case = 63, control = 106), and ADSP admixed (n = 916, case = 340, control = 576). We have updated our manuscript to include this information.

Q. It seems all the results were based on only one set of results. The authors should consider multiple repeats to obtain reliable comparisons. Is DisPred robust to different random seeds, different ancestral admixtures?

Response: We thank the reviewer for providing this constructive feedback. We have considered multiple repeats of our experiments and updated the manuscript with these new experiments. We have found that DisPred consistently performs well when we have different seeds for splitting the datasets and different random seeds for initializing our deep neural networks.

Q. No computational costs and code availability mentioned about DisPred.

Response: We have updated the manuscript with the computational costs and availability of codes for DisPred.

Q. Some literature reviews about previous efforts of deep learning models in polygenic scores would be helpful.

Response: Although the literature around the use of deep learning models for polygenic scores is limited, we have updated our manuscript to include some available works to cover this area of works.

Q. Figure 2 colors by self-reported ancestry, but it may not correctly reflect the genetic ancestry as the lower left panel shows. It will be better to color by genetic ancestry. Also, it seems there are still some clusters based on the colors in the phenotype-specific representations. For example, most of the red dots are on the left and yellow dots on the right in the lower right panel. Any reasonable explanations?

Response: We consider self-reported ancestry labels in the training dataset to train the disentangling autoencoder. Hence, we use the same color in Figure 2 to demonstrate if our disentangling autoencoder correctly separates the ancestry factor. Different disentangling methods, although attempts to separate the underlying factors, may not necessarily achieve the complete disentanglement resulting in the information leakage, and that might have caused the formation of cluster-like structure even in the phenotype-specific representations. However, as the UMAP plot demonstrates, the disentanglement is clear for ancestry-specific representations.

Q. Different ancestry cutoffs for ADSP and UKB doesn't make sense to me. The authors should consider using the same thresholds.

Response: This is primarily because of the differences in these two cohorts. Setting UKB to a threshold similar to the ADSP will provide us with all European individuals because the European population heavily dominates UKB. As such, to find the optimal balance of non-European participants, we used different thresholds.

Q. Figure 4 is not informative. It seems the performance of DisPred on the two panels of the figure are exactly the same. More explanations needed.

Response: In Figure 4, we aim to demonstrate how another method performs when training and test dataset have ancestrally aligned or mismatched, and since DisPred is trained using all the data, the results for the DisPred are the same for both panels.

Reviewer 2

Q. Regarding not using additional covariates in the training.

Response: Since the training dataset and test dataset might have different meaning to the covariates, they may not be meaningful to consider in the study like ours. Deep learning literature, most of the time, works only on the data and instead, argues for learning robust representation to adapt to the covariate shift. Since our primary contribution is a novel deep learning method, we mostly followed the experimental setup in the deep learning literature. However, we acknowledge that not including additional covariates in the training is a limitation of this study and have added it in our discussion section.

Q. In addition to a method that does not induce additional sparsity as comparison.

Response: We have added PRSBayes as a new comparison method that does not induce additional sparsity. This method is based on Naive Bayes approach and we have provided more details in the updated manuscript.

Q. The authors themselves mention that the training set is strongly restricted to significant variants. While I agree that it's fair that all methods are based on the same training data, this is not a realistic approach to compute PRS. Most methods, either based on summary statistics as SBayesR, LDpred2 or lassosum or individual-level-data based as snpnet

(Lasso), would be based on genome-wide data. Can the authors add additional comparisons based on genome-wide data (also for DisPred, if possible)?

Response: We agree that this is the current limitation of our study. The scope of the current work aims to propose a new deep learning approach and its application in AD genetics and thus, we focus on limited variants. However, as reviewer mentioned, we demonstrate that our method, compared to all the standard methods considered in the genetics literature, performs well, especially for the minority population, which essentially what the design of the deep learning approach (i.e., learning disentangled representation) aims for. As such, in future, when we have efficient neural networks that can be easily trained and executed in genome-wide level, we believe the framework like ours could still be favorable as we learn the low-dimensional non-linear structure and use such latent space to remove the underlying population structure making our prediction unbiased and fair.

Q. It would be helpful to show confidence intervals of the AUC (e.g. bootstrapped CI) in Figure 3 to assess the significance of the differences.

Response: We have added robustness analysis, considering multiple repeats of our experiments and updated the manuscript with these new experiments. We have found that DisPred consistently performs well when we have different seeds for splitting the datasets and different random seeds for initializing our deep neural networks.

Q. In Figure 4, it seems that both models, DisPred and PRS-Lasso EUR, perform worse on the target population (EUR) than on the mismatch population (AFR). Can the authors comment on that?

Response: There is a big difference between the sample size of EUR and AFR participants in ADSP dataset, and we believe it's the behavior of the sample that has resulted in this behavior. But we show that our proposed model DisPred performs well even when there is a mismatch between training and target population.

Q. For European data, DisPred does not seem to be the best method, i.e. the increase in prediction performance for non-EUR individuals comes with a decrease in prediction performance for EUR individuals. I would suggest mentioning that point in the discussion.

Response: We have added this point in the discussion section.

Q. In lines 188-191 it is stated that the training set consists completely of ADSP data whereas the UKB data is used for testing purposes only. However, in the methods section “Model architecture and training setting” it seems that the autoencoder was also trained on UKB data. Could the authors please clarify if a separate autoencoder was trained on UKB data?

Response: The results presented in the current study are from autoencoders trained only on the ADSP data, and the writing in the methods section was from the previous versions where we also trained separate autoencoders for UKB. We have fixed this in writing and would like to thank the reviewer for pointing this out.

Q. For the prediction model, as the disease label is binary, I assume it is rather a logistic regression than a linear regression (lines 161-166).

Response: We analyzed linear and logistic regression and didn't find much difference in the result when the disease label is binary. For UKB, since we have a proxy label, if we want to train the model with UKB data, we have to use linear regression. As such, we eventually resorted to using linear regression, considering it will be helpful in scenarios where we don't have an actual label and might have to rely on proxy labels like UKB. For the final AUC calculation, linear regression can still successfully assign class labels based on some threshold on fitted values (in our case, a threshold of 0.5).

Q. In the methods section “Baseline methods” it says “ PRS by clumping (PRS-Clumping) and PRS by Lasso-based penalized regression using summary statistics (PRS-Lasso)” (lines 475-476) but it seems that the Lasso was fitted on the (pre-filtered) individual-level data instead of summary statistics.

Response: This typo has been corrected in the new version.

Reviewer 3

Q. AD was the featured application in this paper but the authors claim that these results can be generalized to other phenotypes without performing simulations under various types of genetic architecture. If the authors do

not plan to add simulations in the revision, at least remove the APOE region (the whole region, not just the 2 SNPs) from the data and see if DisPred still achieves a substantial gain compared to other approaches?

Response: As suggested by the reviewer, we conducted this experiment, where we removed the whole APOE region from our data, eliminating 832 variants. As a result, the performance of our method for ADSP AFR was reduced to 0.535, compared to the Lasso-EUR, the best method in our comparison list, which resulted in 0.522. As shown, although DisPred was able to achieve better results, it is not a huge gain compared to the compared approach.

Q. Is there a reason not to adjust for covariates in the analysis? Even if there are methodological limitations that make it difficult to include covariates during model training, it should be an important to add covariates (e.g., genetic PCs) in model evaluation just to be sure that the gain we see in DisPred is not explained by confounding (e.g., with ancestry composition within each sub population).

Response: Since the training dataset and test dataset might have different meaning to the covariates, they may not be meaningful to consider in the study like ours. Deep learning literature, most of the time, works only on the data and instead, argues for learning robust representation to adapt to the covariate shift. Since our primary contribution is a novel deep learning method, we mostly followed the experimental setup in the deep learning literature. However, we acknowledge that not including additional covariates in the training is a limitation of this study and have added it in our discussion section.

Q. On the inadequacy of the literature review.

Response: We thank the reviewer for pointing out some examples of active research topics in statistical genetics. We have included them in our updated manuscript, along with discussions on how we couldn't do a comprehensive comparison with these existing approaches as our primary focus is the development of deep learning methods.

Q. Related to the comment above, the fact that DisPred requires individual-level data will be viewed as a limitation by many in the field. It doesn't seem straightforward for DisPred to incorporate publicly available GWAS summary statistics (which can be from multiple ancestries) into its

model training procedure. It will be important to mention this in the Discussion section.

Response: We have mentioned this point in the discussion section.

Q. How well does DisPred perform if P_z and P_x (Fig 1c) are evaluated separately? I'm wondering if the gain of performances comes from P_z or from the fact that multiple models are combined through ensemble learning.

Response: We have experimented with P_z alone and found it performs reasonably well. For instance, for admixed individuals as test population ($n = 916$), the top-three best models are: DisPred (0.6157), P_z alone, i.e., Disentgl-AE (0.6105), and PRS-Lasso-EUR (0.5914). As pointed out in the paper, the ensembling helps boost the performance of Disentgl-AE.

Q. Comments on computation time and cost will be helpful.

Response: We have updated the manuscript with the computational costs and availability of codes for DisPred.

Q. Abstract and Intro: I am not sure “confounded by the underlying population structure” is the right way to describe the PRS transferability problem. It is a common view in the field that LD and allele frequency differences, and differences in genetic causal effects (which can be caused by gene-environment interaction for example) are all plausible reasons for low portability of PRS across ancestries. Maybe you could argue MAF difference is a form of “confounding” but certainly some of these other factors aren't.

Response: We aim to present how prediction models are biased towards the training population and have updated the manuscript by avoiding the description as “confounded by the underlying population structure”.

Q. Consider revising the color scheme in Fig 5. It's a bit hard to relate the lines to the legend.

Response: Fig 5 has been revised.

REVIEWERS' COMMENTS:

Reviewer #1 (Remarks to the Author):

The authors have carefully addressed my previous critiques and I have not additional comments.

Reviewer #2 (Remarks to the Author):

I thank the authors for the revised manuscript and their extensive responses. My comments were mostly addressed adequately. However, the authors state that they included as a limitation that they do not include additional covariates but I can't find that limitation in the revised manuscript.